# On the Constrained Time-Series Generation Problem

**Andrea Coletta**
J.P. Morgan AI Research
London, UK

**Sriram Gopalakrishan**
J.P. Morgan AI Research
New York, USA

**Daniel Borrajo**
J.P. Morgan AI Research
Madrid, ESP

**Svitlana Vyetrenko**
J.P. Morgan AI Research
New York, USA

## Abstract

Synthetic time series are often used in practical applications to augment the historical time series dataset, amplify the occurrence of rare events and also create counterfactual scenarios. Distributional-similarity (which we refer to as realism) as well as the satisfaction of certain numerical constraints are common requirements for counterfactual time series generation. For instance, the US Federal Reserve publishes synthetic market stress scenarios given by the constrained time series for financial institutions to assess their performance in hypothetical recessions. Existing approaches for generating constrained time series usually penalize training loss to enforce constraints, and reject non-conforming samples. However, these approaches would require re-training if we change constraints, and rejection sampling can be computationally expensive, or impractical for complex constraints. In this paper, we propose a novel set of methods to tackle the constrained time series generation problem and provide efficient sampling while ensuring the realism of generated time series. In particular, we frame the problem using a constrained optimization framework and then we propose a set of generative methods including "GuidedDiffTime", a guided diffusion model. We empirically evaluate our work on several datasets for financial and energy data, where incorporating constraints is critical. We show that our approaches outperform existing work both qualitatively and quantitatively, and that "GuidedDiffTime" does not require re-training for new constraints, resulting in a significant carbon footprint reduction, up to 92% w.r.t. existing deep learning methods.

## 1 Introduction

In recent years, synthetic time series (TS) have gained popularity in various applications such as data augmentation, forecasting, and imputation of missing values [1, 2, 3, 4, 5]. Additionally, synthetic TS are extremely useful to generate unseen and counterfactual scenarios, where we can test hypotheses and algorithms before employing them in real settings [6]. For example, in financial markets, it can be very useful to test trading strategies on unseen hypothetical markets scenarios, as poorly tested algorithms can lead to large losses for investors, as well as to overall market instability [7, 8]. In order to be useful, such hypothetical market stress scenarios need to be realistic - i.e., the synthetic market TS need to have statistical properties similar to the historical ones. They also need to satisfy certain constraints supplied by experts on how hypothetical market shock scenarios can potentially unfold. For instance, in order to ensure financial market stability, the US Federal Reserve annually assesses the market conditions and publishes a set of constrained market stress scenarios that financial institutions must subject their portfolios to, in order to estimate and adjust for their losses in case of market downturns [9].

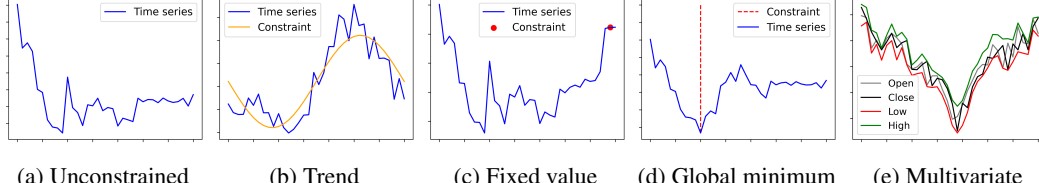

Figure 1: An example of synthetic stock market time-series under different constraints: (a) unconstrained generation; (b) a time-series following a trend constraint; (c) the final value of the TS has to hold a specific value; (d) the global minimum must be at a given time point; (e) multi-variate TS where the *High* and *Low* dimensions have the maximum and minimum values, respectively.

Our work targets the problem of generating constrained TS that are both statistically similar to historical times series and match a given set of constraints. These constraints can be imposed by the underlying physical process that generates the data. For example, synthetic energy data should adhere to the principle of 'energy conservation' [10]. Or, as in the preceding example of the US Federal Reserve stress scenarios, constraints can be used to generate counterfactual synthetic TS with some given conditions, e.g., a stock market index decreases by 5% [9, 11].

**Related work**: Existing work employs deep generative models (DGMs) to capture the statistical data properties and temporal dynamics of TS [1, 12, 13, 14, 15], and additional constraints are usually introduced by penalizing the generative model proportionally to the mass it allocates to invalid data [16, 17, 18]; or by adding a regularization term to the loss function [19, 20]. Other approaches condition the generative process by encoding constraints and feeding them into the model [10]; or they reject and re-sample TS that do not match the constraints [21]. Finally, a different line of work proposes special-purpose solutions, with ad-hoc architectures or sampling methods, which however tackle specific applications (not TS generation) [22, 23, 24, 25]. In general, while most of these models are able to reproduce the real data statistics, complex constraints can still be challenging to be guaranteed. Most importantly, as DGMs incorporate constraints during training, a change to the constraints may require re-training since the learned distribution of a DGM may no longer cover the target distribution, and thus even rejection sampling would not be effective [26].

In this paper, we tackle the constrained TS generation problem for different constraint types, and we present and compare a set of methods that can incorporate constraints into TS generation. First, we present an optimization-based approach in which we compile the problem into a constrained optimization problem (**COP**) [27] and generate synthetic data from an explicit formulation of data properties and constraints. To overcome the need for an explicit definition of data properties, we employ a DGM that can implicitly capture data properties from training data. In particular, we introduce **DiffTime**, an approach based on conditional denoising diffusion models [28] where several constraints can condition the data generation. In addition, we show that any kind of constraint can be applied to diffusion models by penalizing the model proportionally to the constraint violation during training. This approach, called **Loss-DiffTime**, shows good performance with efficient sampling, but requires re-training upon a new constraint. Finally, to increase computational efficiency and reduce the carbon footprint of the model [29], we propose a guided diffusion model **Guided-DiffTime** that does not require re-training upon changes in the constraints —- at inference, it can adjust the TS generation process based on the input constraints.

Our main contributions can be summarized as follows:

- We formally define the constrained TS problem, and characterize different types of constraints for synthetic TS.

- We propose and compare different approaches to generate synthetic TS. We evaluate their advantages and disadvantages with respect to different measures of performance and constraint types. We show how COP can also be used for post-hoc fine-tuning of TS, such that synthetic TS generated by any DGM can be adjusted to guarantee constraint satisfaction.

- We empirically demonstrate that our approaches outperform existing work in many TS domains, both qualitatively and quantitatively. We show that *DiffTime* outperforms existing state-of-art models in unconstrained TS generation, while ***Guided-DiffTime* is the only solution where re-training for new constraints is not necessary, resulting in a significant carbon footprint reduction.**

## 2 Definitions and Problem Formulation

The constrained TS generation problem requires generating synthetic TS data, where each TS is defined in the sample space $\chi = \mathbb{R}^{L \times K}$ where $L$ is the length of the TS and $K$ is the number of features. Our goal is to generate synthetic data such that the synthetic distribution approximates the input data distribution, *and* each TS also conforms to user-specified constraints. The problem input is a tuple $\langle \mathbb{D} = \{\mathbf{x}^i\}_{i=1}^N, C \rangle$ consisting of a dataset $\mathbb{D}$ of $N$ time series $\mathbf{x}^i \in \chi, i \in [1..N]$ and a list of constraints $C$ that a synthetic TS should conform to. The constraints include realism constraints (see Section 3.1). Henceforth, we will drop the sample index for $\mathbf{x}$ unless needed, and only keep the position and feature indices. We also shortly denote $[K] \triangleq \{0, \ldots, K\}$ and $[L] \triangleq \{0, \ldots, L\}$.

Constraints — like those in Figure 1 — are defined as tuples of the form $\langle t, f \rangle \in C$, where $t$ can be either *soft* or *hard* and a differentiable function $f$. If the constraint type is *hard*, then $f$ can be an inequality or an equality constraint. An inequality constraint is of the form $f(\hat{\mathbf{x}}) \leq 0$ where $\hat{\mathbf{x}}$ is the generated synthetic TS. An equality constraint is of the form $f(\hat{\mathbf{x}}) = 0$. Hard constraints are required to hold in a generated TS. Otherwise, the TS is rejected. Soft constraints are of the form $f : \chi \to \mathbb{R}$ whose value we would like to optimize (minimize) for. Therefore, by definition, soft constraints do not require sample rejection. The constraints can be defined with respect to individual synthetic TS samples $\hat{\mathbf{x}}$, or at the dataset level (distribution-related constraints). As a type of soft constraint, we define *trend-lines* (see Figure 1.b) as a time series $\mathbf{s} \in \chi$. This constraint tells a generative method to minimize the L2 distance between the trend and the corresponding points of the synthetic TS. Formally, the synthetic TS $\hat{\mathbf{x}}$ would be optimized as to minimize $\|\mathbf{s} - \hat{\mathbf{x}}\|_2^2$.

Additionally, both soft and hard constraints can be categorized into *local* and *global* constraints. Global constraints are those that compare across all the points in the TS. For example, we can enforce that $x_{i,j} \leq x_{3,0}, \forall(i,j) \in [L] \times [K]$ such that the maximum value is at $x_{3,0}$. Local constraints are those that only refer only to a subset of points. For instance, requiring $(x_{i,j} = 2.5)$ for a given point $(i,j) \in [L] \times [K]$. We refer to this kind of constraint as *fixed-point constraints* (see Figure 1.c) since they require that the value of the TS is fixed at that point to a specific value. The set of all fixed-point constraints is $\mathcal{R}$, where each element $r_{i,j} \in \mathbb{R}$ and $(i,j) \in [L] \times [K]$.

The aforementioned types of constraints are explicit. Additionally, the problem of synthetic data generation requires statistical similarity between the input and the synthetic datasets, which can either be built-in into the data generating method (e.g., by design GANs generate data that is distributionally similar to the input [2]) or specified explicitly as constraints in the model (e.g., autocorrelation similarity can be an explicit constraint). The methods presented herein assume that the constraints are differentiable. This is needed for deep generative methods, such as diffusion models, where constraints need to be incorporated into the training or inference process. If the functions are differentiable, then a straightforward approach [30] to incorporate them into the loss is:

$$loss(\mathbf{x}, \hat{\mathbf{x}}) = objective\_loss(\mathbf{x}, \hat{\mathbf{x}}) + \lambda_g \text{ReLU}(g(\hat{\mathbf{x}})) + \lambda_h h(\hat{\mathbf{x}})^2 \tag{1}$$

where $g(\hat{\mathbf{x}})$ and $h(\hat{\mathbf{x}})$ are the inequality and an equality constraint respectively, which are added as soft constraints into the loss function with penalty terms $\lambda_g$ and $\lambda_h$. However, incorporating constraints into the loss function may not guarantee constraint-conforming solutions, but good candidate or starting solutions that we can then fine-tune (i.e., adjust to guarantee constraints). If the constraints are not differentiable, one can use approaches that compute the loss for such a "rule" using perturbations [10].

## 3 Constrained Time-Series Generation - Proposed Approaches

In this section, we introduce several approaches to tackle the constrained TS generation problem. In particular, we discuss their advantages and disadvantages, and how they handle different scenarios and constraints.

### 3.1 Constrained Optimization Problem (COP)

Our first model tackles the synthetic TS generation problem as a Constrained Optimization Problem (COP) in which we treat each point $x_{i,j} \in \mathbf{x}$ as a decision variable to be optimized. We will refer to this method simply as "*COP-method*". A COP problem is defined by an objective function that a solution is optimized for, and set of constraints that need to be satisfied by the solution.

We can use COP in two ways, as a generative method, and as a fine-tuning method. If COP is used for generating synthetic TS, then we take as input a real sample $\mathbf{x}$ from $\mathbb{D}$ as the starting TS (i.e., seed) for generation, and set the objective to maximize the difference between the seed and the synthetic TS. Formally, we maximize the L2 norm of the TS difference: $obj(\mathbf{x}, \hat{\mathbf{x}}) = \|(\mathbf{x} - \hat{\mathbf{x}})\|_2$, where $\mathbf{x}$ is the seed TS and $\hat{\mathbf{x}}$ is the generated TS. COP can also be used for fine-tuning candidate solutions generated by other methods, such as by a diffusion model. When using COP as a fine-tuning method, the candidate solution generated by the other method becomes the seed TS, and the objective simply changes from maximizing the L2 difference to minimizing it; this is to preserve the information from the candidate TS and just search in the space of nearby solutions for one that satisfies all the constraints (if any failed). This can be helpful in fixing almost correct solutions, rather than using rejection sampling.

The constraints for the COP formulation come from $C$. Additionally, when using COP as a generative process we need to add constraints to satisfy the desired distributional properties that methods like GANs would implicitly handle, such as preserving the distribution of the autocorrelation of returns for stock data. This is done by constraining the COP solver to try and match the desired statistical properties of the seed TS; by matching the property at the sample level, we seek to match the distribution of that property at the dataset level. We do this by computing the target value from the seed TS and comparing it with the corresponding value from the synthetic TS. Specifically, we constrain the COP to limit the magnitude of the error between the property value computed for $\hat{\mathbf{x}}$ and $\mathbf{x}$ within an allowed amount (a budget for error tolerance). This is done by using an inequality constraint as follows: $e(z(\hat{\mathbf{x}}), z(\mathbf{x})) - b \leq 0$, where $b$ is the budget we set ($b = 0.1$ in our experiments), $z(.)$ is the function that computes the desired property, and $e(.)$ is the error function that measures the error between the target and generated values. For example, for autocorrelation of returns in stock data, the TS property is a vector, so the $e(.)$ is the L2-norm of the difference. If the COP solver cannot find a solution within the allowed error tolerance, we double the budget and repeat the process for up to a fixed number of $\eta$ repeats (we set $\eta = 10$ in our experiments). In the supplementary material, we show how distributional constraints can be learned directly from the input data, by training a Wasserstein-GAN [31, 32] and using the GAN *critic* in the objective function.

Once we define the objective function and constraints for COP, we can employ one of the many solvers available to compute the synthetic TS. In our experiments, we use the Sequential Least Squares Programming (SLSQP) solver [27] in Scipy's optimization module [33].

## 3.2 *DiffTime* - Conditional Diffusion Model for Time Series generation

In the previous section, we introduced using COP to generate synthetic TS while guaranteeing the input constraints and data properties. However, such COP problems may be non-linear, and solving a non-linear problem is in general difficult and computationally expensive, especially with multi-variate and long time-series (see Section 4). In this section, we introduce a conditional diffusion model named **DiffTime** that leverages the latest advancements in score-based diffusion models [5, 34, 28, 35] to generate synthetic TS. Our model can generate realistic TS and cope with *Trend* and *Fixed Points* constraints by conditioning the generative model.

**Denoising diffusion models**    Denoising Diffusion models are latent variable models which are trained to generate samples by gradually removing noise — denoising — from samples corrupted by Gaussian noise [28]. These models approximate a real data distribution $q(\mathbf{x}_0)$ by learning a model distribution $p_\theta(\mathbf{x}_0) := \int p_\theta(\mathbf{x}_{0:T}) \, d\mathbf{x}_{1:T}$, where the latent variables $\mathbf{x}_{1:T}$ are defined in the same space $\mathcal{X}$ of the sample $\mathbf{x}_0$. The training follows: a *forward process* that progressively adds noise to the sample $\mathbf{x}_0$; and a *reverse process* where the generative process gradually denoises a noisy observation. The forward process is described with the following Markov chain with Gaussian transitions parameterized by $\beta_{1:T}$:

$$q(\mathbf{x}_{1:T} \mid \mathbf{x}_0) := \prod_{t=1}^{T} q(\mathbf{x}_t \mid \mathbf{x}_{t-1}), \qquad q(\mathbf{x}_t \mid \mathbf{x}_{t-1}) := \mathcal{N}\left(\sqrt{1 - \beta_t}\mathbf{x}_{t-1}, \beta_t \mathbf{I}\right) \qquad (2)$$

It admits the following close form $q(\mathbf{x}_t \mid \mathbf{x}_0) = \mathcal{N}(\mathbf{x}_t; \sqrt{\hat{\alpha}_t}\mathbf{x}_0, (1 - \hat{\alpha}_t)\mathbf{I})$, where $\alpha_t := 1 - \beta_t$ and $\hat{\alpha}_t := \prod_{i=1}^{t} \alpha_i$, which allows sampling $\mathbf{x}_t$ at any arbitrary diffusion step $t$. The generation is

performed by the *reverse process* defined as a Markov Chain starting at $p(\mathbf{x}_T) = \mathcal{N}(\mathbf{x}_T; \mathbf{0}, \mathbf{I})$:

$$p_\theta(\mathbf{x}_{0:T}) := p(\mathbf{x}_T) \prod_{t=1}^{T} p_\theta(\mathbf{x}_{t-1}|\mathbf{x}_t), \qquad p_\theta(\mathbf{x}_{t-1}|\mathbf{x}_t) := \mathcal{N}(\mathbf{x}_{t-1}; \boldsymbol{\mu}_\theta(\mathbf{x}_t, t), \boldsymbol{\Sigma}_\theta(\mathbf{x}_t, t)) \quad (3)$$

Following the formulation of Denoising Diffusion Probabilistic Models (DDPM) [28] we parameterize the *reverse process* as follows:

$$\boldsymbol{\mu}_\theta(\mathbf{x}_t, t) = \frac{1}{\sqrt{\hat{\alpha}_t}} \left( \mathbf{x}_t - \frac{\beta_t}{\sqrt{1 - \hat{\alpha}_t}} \boldsymbol{\epsilon}_\theta(\mathbf{x}_t, t) \right), \quad \boldsymbol{\Sigma}_\theta(\mathbf{x}_t, t) = \sigma^2 \mathbf{I}, \text{ where } \sigma^2 = \sqrt{\beta_t} \quad (4)$$

where $\boldsymbol{\epsilon}_\theta$ is a trainable denoising function that predicts $\boldsymbol{\epsilon}$ from $\mathbf{x}_t$, and the choice of $\beta$ corresponds to the upper bound on the reverse process entropy [35]. This function is approximated through a deep neural network trained according to the following objective:

$$L(\theta) := \mathbb{E}_{t, \mathbf{x}_0, \boldsymbol{\epsilon}} \left\| \boldsymbol{\epsilon} - \boldsymbol{\epsilon}_\theta(\sqrt{\hat{\alpha}_t} \mathbf{x}_0 + \sqrt{1 - \hat{\alpha}_t} \boldsymbol{\epsilon}, t) \right\|^2 \quad (5)$$

where $t$ is uniformly sampled between 1 and $T$, and the noise is Gaussian $\boldsymbol{\epsilon} \sim \mathcal{N}(0, \boldsymbol{I})$. The diffusion steps $T$ and variances $\beta_t$ control the expressiveness of the diffusion process and they are important hyperparameters to guarantee that the forward and reverse processes have the same functional form [35].

**Conditional diffusion models** Our *DiffTime* model – which is a conditional diffusion model — supports both trend and fixed point constraints that were defined in Section 2. To constrain a particular trend, we condition the diffusion process using a trend TS $\mathbf{s} \in \chi$. Following recent work on conditional diffusion models [5], we define the following model distribution:

$$p_\theta(\mathbf{x}_{0:T}|\mathbf{s}) := p(\mathbf{x}_T) \prod_{t=1}^{T} p_\theta(\mathbf{x}_{t-1}|\mathbf{x}_t, \mathbf{s}), \quad p_\theta(\mathbf{x}_{t-1} \mid \mathbf{x}_t, \mathbf{s}) := \mathcal{N}(\mathbf{x}_{t-1}; \boldsymbol{\mu}_\theta(\mathbf{x}_t, t|\mathbf{s}), \boldsymbol{\Sigma}_\theta(\mathbf{x}_t, t|\mathbf{s})). \quad (6)$$

which we learn by extending the parametrization in Eq. 4 with a conditional denoising function $\boldsymbol{\epsilon}_\theta$:

$$\boldsymbol{\mu}_\theta(\mathbf{x}_t, t \mid \mathbf{s}) = \frac{1}{\sqrt{\hat{\alpha}_t}} \left( \mathbf{x}_t - \frac{\beta_t}{\sqrt{1 - \hat{\alpha}_t}} \boldsymbol{\epsilon}_\theta(\mathbf{x}_t, t \mid \mathbf{s}) \right) \quad (7)$$

where the $\boldsymbol{\Sigma}_\theta(\mathbf{x}_t, t \mid \mathbf{s}) = \sigma^2 \mathbf{I}$. In this formulation, the trend is provided during each diffusion step $t$, without any noise added to the conditioning trend. During the training, we extract the trend $\mathbf{s}$ directly from the input TS $\mathbf{x}_0$, which can be a simple linear or polynomial interpolation; during inference, the trend can be defined by the user at inference time. We recall that this is a *soft constraint*, meaning that we do not expect the generated TS to exactly retrace the trend. In particular, during training, we provide a trend that is a low-order polynomial approximation of $\mathbf{x}_0$ to avoid the model from copying the trend $\mathbf{s}$. Figure 1.b shows an example of the trend constraint.

Thus, the *DiffTime* training procedure minimizes the following revised loss function:

$$L(\theta) := \mathbb{E}_{t, \mathbf{x}_0, \boldsymbol{\epsilon}} \left\| \boldsymbol{\epsilon} - \boldsymbol{\epsilon}_\theta(\sqrt{\hat{\alpha}_t} \mathbf{x}_0 + \sqrt{1 - \hat{\alpha}_t} \boldsymbol{\epsilon}, t \mid \mathbf{s}) \right\|^2 \quad (8)$$

**Fixed Points.** To satisfy the *fixed point* constraints, which are hard constraints, we modify the *reverse process* of *DiffTime* to explicitly include them in the latent variables $\mathbf{x}_{1:T}$. We recall that $\mathcal{R}$ is the set of fixed point constraints, such that a fixed point constraint $r_{i,j} \in \mathcal{R}$ with $(i, j) \in [L] \times [K]$. Thus, at each diffusion step $t$ we explicitly enforce the fixed-points values in the noisy time-series $\mathbf{x}_t$, such that $\forall r_{i,j} \in \mathcal{R}, x_{i,j} = r_{i,j}$ where $x_{i,j} \in \mathbf{x}_t$. This approach would guarantee that the generated TS have the desired fixed-point values. Most importantly, we experimentally validated that the forward process generates consistent neighboring points (around the constrained fixed-points) which means that the synthetic samples are conditioned by the fixed points, and preserve the realism of the original input data. During training, we *randomly* sample the *fixed points* from the input TS($\mathbf{x}_0$) and require the diffusion process to conform to those fixed points. At inference, the *fixed points* can be provided by the user. Figure 1.c shows an example of a fixed point at the end of the TS, where the TS adapts to deal with the fixed point.

In the supplementary material, we provide additional details, network architecture, and the algorithm pseudo-codes.

### 3.3 *Loss-DiffTime* - Constrained generation with diffusion models

In *DiffTime*, we leverage conditional diffusion models to support trend and fixed values for generating TS. However, just by conditioning the model generation is not possible to encode all the constraints. A common solution is to penalize the generative model proportionally to how much the generated TS violates the input constraint [36].

In this section, we propose *Loss-DiffTime* where a *constraint penalty* is applied to deal with any kind of constraint. The penalty function $f_c : \mathcal{X} \to \mathbb{R}$ is added to the learning objective of the diffusion model, and it evaluates whether the generated TS $\hat{\mathbf{x}}$ meets the input constraint. We discuss the penalty function $f_c$ for constraints in Section 2 and in Equation 1. With $f_c$ in the loss, the greater the constraint violation is, the greater the model loss during training will be. However, the optimization problem in Eq. 5 predicts the noise component for the sample $\mathbf{x}_0$, into which we cannot directly feed to our penalty function. Moreover, we cannot apply $f_c(\mathbf{x})$ to a noisy sample $\mathbf{x}_t$ as the constraints may be evaluated only on the final sample. Therefore, to apply our penalty function, we re-parametrize the optimization problem and force the diffusion model to explicitly model the final sample $\hat{\mathbf{x}}_0$ at every step as follows:

$$L(\theta) := \mathbb{E}_{t,\mathbf{x}_0,\boldsymbol{\epsilon}} \left[ \|\boldsymbol{\epsilon} - \boldsymbol{\epsilon}_\theta(\mathbf{x}_t, t \mid s)\|^2 + \rho f_c(\hat{\mathbf{x}}_0) \right] \tag{9}$$

where $\mathbf{x}_t = \sqrt{\hat{\alpha}_t}\mathbf{x}_0 + \sqrt{1 - \hat{\alpha}_t}\boldsymbol{\epsilon}$ and $\hat{\mathbf{x}}_0 = \frac{1}{\sqrt{\alpha_t}}\left(\mathbf{x}_t - \frac{1-\alpha_t}{\sqrt{1-\hat{\alpha}_t}}\boldsymbol{\epsilon}_\theta(\mathbf{x}_t, t)\right)$. We consider that any constraint in $C$ can be differentiable (as discussed in Section 2). So, we can train our diffusion model following Eq. 9 where $\rho$ is a scale parameter used to adjust the importance of the constraint loss. The conditional information of the trend $\mathbf{s}$ can be removed if we do not need to enforce any trend constraint. Figure 1.d and Figure 1.e show two examples of more complex constraints with *Loss-DiffTime*.

### 3.4 *Guided-DiffTime* - Guided Diffusion models for constrained generation

The *Loss-DiffTime* model is now able to generate real TS while dealing with any constraint. However, we notice two major drawbacks: 1) since we translate constraints to penalty terms in the loss, we need to re-train the model for new constraints; and 2) the diffusion models usually require several iterative steps $T$ which can make it slower and expensive for TS generation. Our final proposed approach, namely *Guided-DiffTime*, solves these two problems and can dramatically reduce the carbon footprint when using DGM for constrained TS generation. In particular, it adopts a Denoising Diffusion Implicit Model (DDIM) [37] which requires fewer diffusion steps at inference. Moreover, by following the groundbreaking work of [38, 39], which shows how to guide a diffusion model using a noisy classifier, we demonstrate how a pre-trained diffusion model can be guided (conditioned) using gradients from differentiable constraints.

DDIM is a class of non-Markovian diffusion processes with the same training objective of classic DDPMs [28], but fewer diffusion steps to generate high-quality samples. In particular, DDIMs keep the same training procedure as DDPMs defined in Section 3.2 while the sampling can be accelerated by using the following re-parametrization of the *reverse process*:

$$\mathbf{x}_{t-1} = \sqrt{\hat{\alpha}_{t-1}}\left(\frac{\mathbf{x}_t - \sqrt{1 - \hat{\alpha}_t} \cdot \epsilon_\theta(\mathbf{x}_t, t)}{\sqrt{\hat{\alpha}_t}}\right) + \sqrt{1 - \hat{\alpha}_{t-1} - \sigma_t^2} \cdot \epsilon_\theta(\mathbf{x}_t, t) + \sigma_t\epsilon \tag{10}$$

where $\hat{\alpha}_0 := 1$ and different parametrizations of $\sigma_t$ lead to different generative processes. We set $\sigma_t = 0, \forall t \in [0, T]$ to have a deterministic forward process from latent variables to the sample $\mathbf{x}_0$ (since the noise term $\sigma_t\epsilon$ is zeroed out). This deterministic forward process defines the DDIM which can use fewer diffusion steps to generate realistic samples. This diffusion steps are defined by a sequence $\tau$ of length $V$ which is a sub-sequence of $[1, \ldots, T]$ with the last value as $T$, i.e., $\tau_V = T$ [37]. For example, $\tau = [1, 4, 9, ..., T]$. Moreover, this parametrization is a generalization of DDPM as setting $\sigma_t = \sqrt{(1 - \alpha_{t-1})/(1 - \alpha_t)}\sqrt{1 - \alpha_t/\alpha_{t-1}}$ describes the original DDPM [37] and the DDIM work showed that re-training of the DDPM model is unnecessary when we change the value of $\sigma$ or the diffusion steps $\tau$.

Given the DDIM, we can then apply the recent results from guided diffusion models [38, 39] to condition each sampling step with the information given by the gradients of the differentiable constraint $f_c$ (see Section 3.3). Algorithm 1 shows the sampling procedure which computes the

**Algorithm 1** *Guided-DiffTime*

---

Input: differentiable constraint $f_c : \mathcal{X} \to \mathbb{R}$, scale parameter $\rho$
Output: new TS, $\mathbf{x}_0$
$\mathbf{x}_T \leftarrow$ sample from $\mathcal{N}(0, \mathbf{I})$
**for all** $t$ from $T$ to 1 **do**
    $\hat{\epsilon} \leftarrow \epsilon_\theta(\mathbf{x}_t, t)$
    $\hat{\epsilon} \leftarrow \hat{\epsilon} - \rho\sqrt{1 - \hat{\alpha}_t} \, \nabla_{\mathbf{x}_t} f_c\big(\frac{1}{\sqrt{\hat{\alpha}_t}}(\mathbf{x}_t - \hat{\epsilon}\sqrt{1 - \hat{\alpha}_t})\big)$
    $\mathbf{x}_{t-1} \leftarrow \sqrt{\hat{\alpha}_{t-1}} \left( \frac{\mathbf{x}_t - \sqrt{1 - \hat{\alpha}_t}\hat{\epsilon}}{\sqrt{\hat{\alpha}_t}} \right) + \sqrt{1 - \hat{\alpha}_{t-1}}\hat{\epsilon}$
**end for**
**return** $\mathbf{x}_0$

---

gradients w.r.t. to the input TS $\mathbf{x}_t$. We recall that the constraint is applied on the final sample $\hat{\mathbf{x}}_0$, computed according to the DDIM reverse process. Again, this approach does not require re-training of the original diffusion model to deal with new constraints, which can be applied just at inference time. Hence, we reduce the carbon footprint of the model, and get a faster time-series generation.

## 4 Experiments

In this section we evaluate our approaches, showing their advantages and disadvantages when applied to different domains and constraints. In particular, we follow the five scenarios shown in Figure 1 while considering multiple real-world and synthetic datasets. For COP we use a subset of the original TS as starting solution (seed), we leave in the supplementary material the analysis of different seeds.

**Baselines** We compare our approaches against existing TS generative models, including GT-GAN[12], TimeGAN [1], RCGAN [40], C-RNN-GAN [41], a Recurrent Neural Networks (RNN) [1] trained with T-Forcing and P-Forcing [42, 43], WaveNET [44], and WaveGAN [45]. For the constrained scenarios, we extend the benchmark architectures to cope with constraints, by introducing a penalty loss [18, 16] or by conditioning the generation process. We also employ *rejection-sampling* and *fine-tuning* with COP on their generated synthetic TS.

**Datasets** We consider three datasets with different characteristics such as periodicity, noise, correlation, and number of features: **1) daily stocks** which uses daily historical Google stock data from 2004 to 2019 with *open, high, low, close, adjusted close, and volume* features [1]; **2) energy data** from the UCI Appliances energy prediction dataset [46] containing 28 features with noisy periodicity and correlation; **3) sines** a synthetic multivariate sinusoidal TS with different frequencies and phases [1].

**Evaluation metrics** For each experimental scenario, we evaluate the generative models and TS along different quantitative and qualitative dimensions. First, we evaluate the **realism** through a *discriminative score* [1], which measures how much the generated samples resemble (i.e., are indistinguishable from) the real data using a post-hoc RNN trained to distinguish between real and generated samples. We evaluate the **distributional-similarity** between the synthetic data and real data by applying t-SNE [47] on both real and synthetic samples; t-SNE shows (in a 2-dimensional space) how well the synthetic distribution covers the original input distribution. Then, we evaluate the **usefulness** of generated samples — how the synthetic data supports a downstream task such as prediction —- by training an RNN on synthetic data and testing its prediction performance on real data (i.e., *predictive-score* [1]). To evaluate how the model satisfies different constraints, we introduce the following metrics: **Perc. error distance** which measures how much the synthetic data follows a trend constraint by evaluating the L2 distance between the TS and the trend; **satisfaction rate** which measures the percentage of time a synthetic TS meets the input constraints; the **inference time** measured as the average seconds required to generate a new sample with a given constraint; and finally the **fine-tuning time** which is the average time, in seconds, needed to enforce constraints over a generated sample, using COP to fine-tune it.
We provide further experiments, including details on the baselines, datasets, metrics, and algorithm hyperparameters in the supplementary material.

## 4.1 Unconstrained Generation

First, we compare the ability of *DiffTime* and *COP* to generate unconstrained TS against existing benchmark datasets and algorithms. In Figure 2 we evaluate the realism with respect to the distributional-similarity, where red dots represent the original TS and blue dots the generated TS. The figure shows that our approaches have significantly better performance with better overlap between red and blue samples.[1]

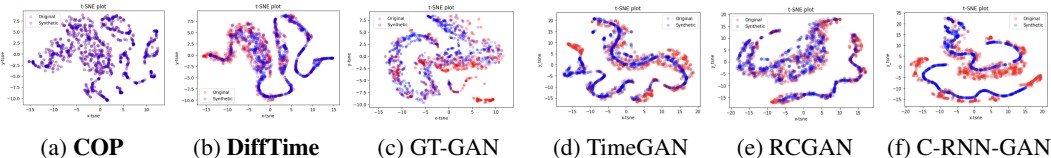

|        (a) **COP**        |       (b) **DiffTime**       |       (c) GT-GAN       |       (d) TimeGAN       |       (e) RCGAN       |       (f) C-RNN-GAN       |

Figure 2: t-SNE visualizations on multivariate stock data, where a greater overlap of blue and red dots shows a better distributional-similarity between the generated data and original data. Our approaches show the best performance.

In Table 1 we measure the *usefulness* and *realism* through the predictive and discriminative scores, respectively. *DiffTime* consistently generates the most useful data with the best predictive score for both Sines and Stocks datasets, while keeping remarkable realism (i.e., discriminative score). *COP* generates excellent synthetic samples as well, indistinguishable from real data with the best discriminative score for both Stocks and Energy. However, we acknowledge COP is advantaged by the original time series as an input seed.

Table 1: Unconstrained Time-Series Generation (Bold indicates best performance).

| Metric | Method | Sines | Stocks | Energy |
|---|---|---|---|---|
| Discriminative Score (Lower the Better) | DiffTime (Ours) | $.013 \pm .006$ | $.097 \pm .016$ | $.445 \pm .004$ |
| | COP (Ours) | $.020 \pm .001$ | $\mathbf{.050 \pm .017}$ | $\mathbf{.101 \pm .019}$ |
| | GT-GAN | $.012 \pm .014$ | $.077 \pm .031$ | $.221 \pm .068$ |
| | TimeGAN | $\mathbf{.011 \pm .008}$ | $.102 \pm .021$ | $.236 \pm .012$ |
| | RCGAN | $.022 \pm .008$ | $.196 \pm .027$ | $.336 \pm .017$ |
| | C-RNN-GAN | $.229 \pm .040$ | $.399 \pm .028$ | $.449 \pm .001$ |
| | T-Forcing | $.495 \pm .001$ | $.226 \pm .035$ | $.483 \pm .004$ |
| | P-Forcing | $.430 \pm .227$ | $.257 \pm .026$ | $.412 \pm .006$ |
| | WaveNet | $.158 \pm .011$ | $.232 \pm .028$ | $.397 \pm .010$ |
| | WaveGAN | $.277 \pm .013$ | $.217 \pm .022$ | $.363 \pm .012$ |
| Predictive Score (Lower the Better) | DiffTime (Ours) | $\mathbf{.093 \pm .000}$ | $\mathbf{.038 \pm .001}$ | $.252 \pm .000$ |
| | COP (Ours) | $.095 \pm .002$ | $.041 \pm .001$ | $\mathbf{.250 \pm .003}$ |
| | GT-GAN | $.097 \pm .000$ | $.040 \pm .000$ | $.312 \pm .002$ |
| | TimeGAN | $.093 \pm .019$ | $\mathbf{.038 \pm .001}$ | $.273 \pm .004$ |
| | RCGAN | $.097 \pm .001$ | $.040 \pm .001$ | $.292 \pm .004$ |
| | C-RNN-GAN | $.127 \pm .004$ | $\mathbf{.038 \pm .000}$ | $.483 \pm .005$ |
| | T-Forcing | $.150 \pm .022$ | $\mathbf{.038 \pm .001}$ | $.315 \pm .005$ |
| | P-Forcing | $.116 \pm .004$ | $.043 \pm .001$ | $.303 \pm .005$ |
| | WaveNet | $.117 \pm .008$ | $.042 \pm .001$ | $.311 \pm .006$ |
| | WaveGAN | $.134 \pm .013$ | $.041 \pm .001$ | $.307 \pm .007$ |
| | Original | $.094 \pm .001$ | $.036 \pm .001$ | $.250 \pm .003$ |

## 4.2 Constrained Generation

We now evaluate the performance of our approaches against the constraints shown in Figure 1 using daily stock data. For univariate constraints (i.e., trend, fixed values, and global minimum) we consider only the *Open* value from the daily stock dataset. We consider as benchmarks the best three SoA approaches from Table 1 (i.e., GT-GAN, TimeGAN, and RCGAN). For trend- and fixed-values constraints we condition their generative process so that different trends and values can be used at inference time. For the other constraints, we add a penalty term in the training loss [18, 16].

---

[1]We report only the top 6 models, leaving the full evaluation to the Supplementary material.

**Soft Constraints** In table 2 we constrain the synthetic TS to follow a given trend, computed as a 3-degree polynomial approximation from the original samples. Our approaches generate synthetic data that are closer to the input trend, with the smallest relative distance (i.e., *perc. error distance*). Moreover, our approaches are among the best in terms of realism and usefulness. In the supplementary material we investigate the use of sinusoidal trends, including additional evaluation metrics.

Table 2: Soft Constraints (Trend) Time-Series Generation (Bold indicates best performance).

| Algo | Discr-Score | Pred-Score | Inference-Time | Perc. Error Distance |
|---|---|---|---|---|
| COP (Ours) | **0.01±0.01** | **0.20±0.00** | 0.73±0.05 | **0.015±0** |
| DiffTime (Ours) | **0.01±0.01** | **0.20±0.00** | 0.02±0.00 | 0.018±0 |
| GT-GAN | 0.04±0.03 | 0.22±0.00 | **0.00±0.00** | 1.378±2 |
| TimeGAN | 0.02±0.02 | **0.20±0.00** | **0.00±0.00** | 0.073±0 |
| RCGAN | 0.02±0.01 | **0.20±0.00** | **0.00±0.00** | 0.071±0 |

**Hard Constraints** In Table 3 we evaluate all the approaches against hard constraints (see Fixed Points, Global Min, and Multivariate in Figure 1). For *Global Min* almost all approaches have a great *satisfaction rate*. However, our approaches are above $0.90$ *and* have the best discriminative and predictive score. Additionally, while $100\%$ of the synthetic time-series generated by TimeGAN and RCGAN guarantee this type of constraint, they do not approximate the input distribution well(see Figure 3). For most complex constraints like the multivariate one, the satisfaction rate drops for most of the benchmarks while for our *GuidedDiffTime* and *COP* the satisfaction rate is still very high, with great realism and usefulness. Finally, when we employ the fixed point constraints, we fix the values of the points at index 6 and 18. All the benchmarks fail to satisfy these constraints, while we show instead that *DiffTime* is able to always guarantee this constraint, by enforcing it during the diffusion steps. Most importantly, it achieves very good discriminative and predictive scores with low inference time. To summarize our results: COP achieves almost always the best realism and usefulness scores, but with higher inference time and using original input TS as seed (which makes the generated TS very similar to the input data); diffusion models are also very powerful with lower inference time and use random noise as input seed as opposed to a real TS (this gives us better variety in TS compared to COP); and *GuidedDiffTime* is able to enforce new constraints without any re-training yet achieving excellent performance.

Table 3: Hard Constraints Time-Series Generation (Bold indicates best performance).

| Constraint | Algo | Discr-Score | Pred-Score | Inference-Time | Satisfaction Rate | Fine-Tuning Time |
|---|---|---|---|---|---|---|
| | COP (Ours) | **0.02±0.01** | **0.20±0.00** | 19.1±1.01 | **1.00±0.00** | **0.00±0.00** |
| | GuidedDiffTime (Ours) | 0.03±0.02 | 0.21±0.00 | 0.03±0.00 | 0.90±0.01 | 3.01±0.10 |
| | LossDiffTime (Ours) | 0.22±0.03 | 0.38±0.00 | 0.02±0.00 | 0.99±0.00 | 6.00±0.60 |
| Global Min | GT-GAN | 0.04±0.02 | 0.22±0.00 | **0.00±0.00** | 0.87±0.02 | 9.30±1.30 |
| | TimeGAN | 0.03±0.02 | 0.21±0.00 | **0.00±0.00** | **1.00±0.00** | **0.00±0.00** |
| | RCGAN | 0.23±0.03 | **0.20±0.00** | **0.00±0.00** | **1.00±0.00** | **0.00±0.00** |
| | COP (Ours) | **0.04±0.02** | **0.04±0.00** | 2.17±0.10 | **1.00±0.00** | **0.00±0.00** |
| | GuidedDiffTime (Ours) | 0.08±0.00 | **0.04±0.10** | 0.15±0.00 | 0.72±0.02 | 31.0±1.50 |
| | LossDiffTime (Ours) | 0.35±0.04 | **0.04±0.01** | 0.14±0.00 | 0.69±0.01 | 57.5±5.01 |
| Multivariate (OHLC) | GT-GAN | 0.22±0.07 | 0.05±0.00 | **0.00±0.00** | 0.05±0.01 | 44.5±3.01 |
| | TimeGAN | 0.24±0.03 | 0.05±0.00 | **0.00±0.00** | 0.51±0.02 | 16.1±1.30 |
| | RCGAN | 0.35±0.04 | **0.04±0.00** | **0.00±0.00** | 0.00±0.00 | 95.1±4.03 |
| | COP (Ours) | **0.02±0.02** | **0.20±0.00** | 0.56±0.11 | **1.00±0.00** | **0.00±0.00** |
| | DiffTime (Ours) | 0.04±0.03 | 0.21±0.00 | 0.01±0.00 | **1.00±0.00** | **0.00±0.00** |
| Two Fixed Points | GT-GAN | 0.04±0.03 | 0.21±0.00 | **0.00±0.00** | 0.00±0.00 | 0.99±0.10 |
| | TimeGAN | 0.03±0.01 | **0.20±0.00** | **0.00±0.00** | 0.00±0.00 | 0.84±0.00 |
| | RCGAN | **0.02±0.02** | **0.20±0.00** | **0.00±0.00** | 0.00±0.00 | 0.87±0.20 |

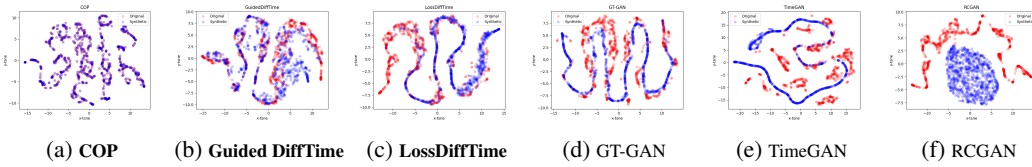

(a) **COP**    (b) **Guided DiffTime**    (c) **LossDiffTime**    (d) GT-GAN    (e) TimeGAN    (f) RCGAN

Figure 3: t-SNE visualizations on *Global-Min* constrained data, where a greater overlap of blue and red dots implies a better distributional-similarity between the generated data and original data. Our approaches show the best performance.

## 5 Conclusions

In summary, we defined the problem of generating synthetic TS data with both soft and hard constraints, and we presented a set of novel methods. We evaluated our approaches on different datasets and we compared their performance against existing state-of-art methods. We showed that our approaches outperform existing work both qualitatively and quantitatively. Most importantly, we introduced *GuidedDiffTime* to handle new constraints without re-training, and we showed that the *COP-method* can be used to fine-tune candidate solutions. Please refer to the supplementary material for more details on experiments comparing the methods presented herein.

## Disclaimer

This paper was prepared for informational purposes by the Artificial Intelligence Research group of JPMorgan Chase & Coànd its affiliates ("JP Morgan"), and is not a product of the Research Department of JP Morgan. JP Morgan makes no representation and warranty whatsoever and disclaims all liability, for the completeness, accuracy or reliability of the information contained herein. This document is not intended as investment research or investment advice, or a recommendation, offer or solicitation for the purchase or sale of any security, financial instrument, financial product or service, or to be used in any way for evaluating the merits of participating in any transaction, and shall not constitute a solicitation under any jurisdiction or to any person, if such solicitation under such jurisdiction or to such person would be unlawful.

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
