# OpenReview forum: "On the Constrained Time-Series Generation Problem"
_NeurIPS.cc/2023/Conference — NeurIPS 2023 poster_

### Official Review · Reviewer_5edF · 2023-06-30

**Soundness:** 3 good
**Presentation:** 4 excellent
**Contribution:** 3 good
**Rating:** 8
**Confidence:** 3

**Summary:**

The authors propose a set of methods for generating synthetic time series, with or without constraints. They defined the different types of constraints possible, including soft and hard constraints.  They developed a COP approach for generating time series from a seed, then presented ways of adding constraints, and delivered an all-inclusive method based on diffusion models: GuidedDiffTime.

**Strengths:**

The paper is very clear and pleasant to read. Everything is very well explained, including the context, the issues and the solutions implemented. The content is substantial and shows a lot of hard work on the part of the authors. The results have been tested on several databases and show the advantages and disadvantages of their solutions.

**Weaknesses:**

Nothing in particular.

**Questions:**

1) What kind of constraints are not included in those you describe? How would you go about adding them and taking them into account?
2) Have you thought about less "traditional" time series that look more like "functions of time"? More precisely, how would your method adapt to very different curve shapes?

**Limitations:**

See questions.

---

> ### Author Rebuttal · Authors · 2023-08-09
>
>
> We thank the reviewer for the positive and constructive feedback. We are happy to add more discussion on the type of constraints and time-series we currently support, and to extend our work in the future, following the reviewer suggestion.
>
>
> **Q.1**  Currently our work cannot cover constraints that involve logical functions over constraints and are not differentiable. For example, we cannot support exclusive-or constraints like ``either $f(x) \leq  3$ or $g(x) \leq 4.5$ but not both". In fact, we cannot model such constraints to be provided as input to the generative models. We plan to address such limitation in future work.
>
> **Q.2** We thank the reviewer for the interesting comment. We currently have one experiment in which we consider a sine-wave dataset, as an example of time series that is a function of time. Some results associated to this can be found in Table 1 of main paper. Nevertheless, we acknowledge that in such examples we considered discrete time-series, i.e., we take observation from the sine function at specific regular times. Considering the different modelling challenges posed by continous functions, we plan to consider this scenarios for future work. Similarly, we will better investigate time-series with changing curve shapes (e.g., a time series that can be represented by a piece-wise continuous trajectory with a distributional shift occurring), in the future, including examples from existing work [11].
>
>
> **References in the global rebuttal comment**

---

> > ### Comment · Reviewer_5edF · 2023-08-20
> >
> > I'd like to thank the authors for their interesting reply. Taking into account the other reviews and the answers given by the authors, I stand by my first rating.

---

### Official Review · Reviewer_DrNe · 2023-07-05

**Soundness:** 3 good
**Presentation:** 4 excellent
**Contribution:** 2 fair
**Rating:** 7
**Confidence:** 3

**Summary:**

The paper proposes a model for generating synthetic time series data given user-specified constraints. The constraints can be either soft (e.g. following trend) or hard ((in)equalility). The first proposed method is framed as a Constrained Optimization Problem (COP), it inputs a seed time series which is either real or sythetic and maximizes or minimizes the L2 distance, respectively, while satisfying the constraints.
The second method, called DiffTime, is a conditional diffusion model -- where the soft constraint (trend) is conditioning the diffusion process. The hard constraint (fixed point) is incorporated in the reverse process.
Thirdly, general constraints are included in the diffusion loss s.t. the model reproduces the sample x_0 with all the constraints satisfied, at every diffusion step.
Finally, sampling from a diffusion-based model can be guided with gradients that come from the constraints directly.

**Strengths:**

The paper is well written. Approaches are straightforward and seem valid. Each extension in Section 3 comes naturally as an answer to a weakness of the previous approach. There is some novelty in the presented methods.

**Weaknesses:**

The originality of most of the approaches is somewhat limited. The first three approaches seem like simple baselines that are applying existing methods to constrained generation. Only the last approach with guidance cannot be directly applied to a GAN or some other generative model.

Motivation to solve this problem. Although the paper mentions reasons to perform constrained generation, the experiments do not support this. A case study or showing benefits on a downstream task would significantly improve the argument.

Conserning the results, in Figure 2 and 3 t-SNE is not very convinving, and in general using t-SNE is not very rigorous. GT-GAN seems to be outperforming DiffTime in Table 1 (upper part).
An interesting baseline would be a naive approach that adds noise to data samples or mixes time series or does some other simple transformation. This would show how much is relative difference between methods significant.
The experiments would benefit from having more datasets and baselines.

**Questions:**

- In COP-method, is L2 objective the best proxy for obtaining diversity/similarity? Could simple pointwise difference guarantee these properties?

- Section 4.1, how is COP exactly "trained"? As pointed in this section, since it's using real data seed, the comparison is not really fair.

- Can you guarantee that the final samples follow true (conditional) data distribution? I imagine some of the proposed methods can create samples that satisfy constraints and are unrelated to data.

**Limitations:**

Performance limitations are discussed.

---

> ### Author Rebuttal · Authors · 2023-08-09
>
> We thank the reviewer for the positive and constructive feedback. We are happy to revise and improve the paper according to the reviewer's comments, and to better motivate our work and its novelty.
>
> ## Weaknesses
>
> **Q.1**
> Our paper presents a set of methods for incorporating constraints during time-series (TS) generation, which is not directly addressed in the current literature.  The first approaches are indeed adaptations of existing models to provide them the ability of constrained TS generation. However, we believe these approaches provide the reader with a more natural narrative and motivation for our proposed new methods, showing how each method address weakness of the previous one. Moreover, some of these approaches achieve remarkable performance, and can be used for specific constraints or tasks.
>
> **Q.2** In the revised version we will add more examples of downstream tasks that can benefit from constrained synthetic TS. We recall that constrained TS generation allows to:
> - augment training datasets by amplifying the occurrence of rare events;
> - test algorithms/models on counterfactual scenarios.
>
> **Financial Market Example:** An example of a downstream task is the testing of existing trading or portfolio strategies from the literature [13,14] for specific stock market scenarios. Ideally, we would like to know how such strategies behave during flash-crashes [9] or market shocks [10, 12]. However, as such events occur rarely, there is shortage of data for such market regimes. Two recent examples are the flash crash of E-MINI SP500 futures [9] on 6th May 2010, or the COVID market crash on March 2020 [10]. While we do not know when and how such rare events will happen, we can use constrained TS generation to create thousand of possible market crashes, with different magnitude and duration, to evaluate the strategies.
> Figure 2 of the attached rebuttal PDF shows that we can generate synthetic market scenarios with similar crashes to E-MINI and COVID. It is worth to notice that, the real market crash data are not used during the diffusion model training, but we can still generate realistic crash scenarios with a specific *constraint* (i.e., from 10\% to 30\% price drops).
> The synthetic market data can be used to test and evaluate such strategies, and study their behaviors when similar events actually happen.
>
> **Q.3** We thank the reviewer for suggesting new baselines for evaluating synthetic TS. We introduced the three new baselines that (1) add Gaussian noise to data samples; (2) mix (average) existing TS; (3) create a simple Brownian motion starting from a real TS. We report quantitative metrics in the table below, while t-SNE plots are shown in Figure 8 of the PDF rebuttal. It is interesting to note that only the mixed TS approach shows good distributional similarity, as averaging two real TS can preserve some distribution information. However, as we report in the Table, such baseline is not competitive with our methods.
>
> $
> \begin{array}{l|lll}
> \hline
> \text{Algo} &  \text{Discr-Score} & \text{Pred-Score} \\\\ \hline
> \text{COP (Ours)} & .050 \pm .017 & .041 \pm .001   \\\\
> \text{DiffTime (Ours)} & .097 \pm .016 & .038 \pm .001  \\\\
> \text{Noised TS} & .491 \pm .091 &  .038 \pm .001 \\\\
> \text{Mixed TS} & .186 \pm .010 &  .041 \pm .001  \\\\
> \text{Brownian TS} &  .085 \pm. 010 &  .038 \pm .000  \\\\
> \hline
> \end{array}
> $
>
>
> ## Questions
>
> **Q1** We agree with the reviewer that L2 distance may not necessarily be the best proxy for diversity, and that we can use any other distance-based metric. We will better clarify this in the paper, providing additional experiments and comparing different distance metrics. In particular, following the reviewer suggestion, we empirically evaluated L2 distance and point-wise distance. Figure 5 in PDF rebuttal shows that both the distance metrics preserve distributional similarity in the synthetic data, which we empirically evaluated using t-SNE. However, the L2 distance achieves better quantitative results, a shown below:
>
> $
> \begin{array}{l|ll}
> \hline
> \text{Algo} &  \text{Discr-Score} & \text{Pred-Score} \\\\ \hline
>  \text{COP L2-distance} & 0.017 \pm 0.006 & 0.203 \pm 0.001 \\\\ \hline
>  \text{COP Pointwise-distance} & 0.021 \pm 0.012 & 0.203 \pm 0.002  \\\\ \hline
>  \end{array}
> $
>
> **Q.2**
> COP frames the task of generating a TS sample as optimizing the value of a set of ordered points that make up the TS sample, such that the sample satisfies domain properties (like auto-correlation). In particular, it starts from an initial sample TS (taken from the dataset, or randomly generated) and optimizes its values according to its objective (e.g., maximize the distance between the generated and initial sample), while respecting some constraints (be them statistical properties or additional structural constraints). This optimization approach avoids training, but usually involves longer inference times. While deep models use the real training set to learn the data distribution, COP uses the training set differently, as starting samples to optimize. Nevertheless, in the supplementary material (Section G.3) we show that COP achieves similar performance when the initial TS samples are generated by random Brownian motion, or by noising original data.
>
> Finally, we would like to highlight that our adaptation of COP for TS generation easily supports complex constraints, and the generation procedure is straightforward. On the other hand, our diffusion models can provide more sophisticated data generation by learning implicit patterns from data, as well as faster sampling.
>
> **Q.3**
> Currently, we do not have any formal guarantee that the synthetic data follows the true conditional distribution, but we can only empirically evaluate that. We agree that this is an interesting discussion, and we will focus on this aspect in a future work.
>
> **References in the global rebuttal comment**

---

> > ### Comment · Reviewer_DrNe · 2023-08-21
> >
> > Thank you for your response. I think some issues still remain but the additional results and clarifications are good enough to justify acceptance. I am raising my score accordingly.

---

### Official Review · Reviewer_DGpF · 2023-07-06

**Soundness:** 3 good
**Presentation:** 2 fair
**Contribution:** 3 good
**Rating:** 5
**Confidence:** 2

**Summary:**

This paper proposes multiple approaches, including a Constrained Optimization Problem (COP) method and a conditional diffusion model called DiffTime, to cope with the problem of generating constrained time series data that adheres to statistical properties and specific user-defined constraints. It also provides a comparative evaluation against existing time series generative models using real-world and synthetic datasets.

**Strengths:**

This paper provides a strong motivation for studying the constrained time series generation problem by highlighting its practical applications and the need for constraint-compliant synthetic data. Additionally, it offers comprehensive experiments against existing models using real-world and synthetic datasets.

**Weaknesses:**

1. The clarity and definition of certain terms in the paper require improvement. For example, terms like "seed" in line 133 and "desired property" in line 145 need clear definitions to enhance understanding. Additionally, in line 154, what does “TS property is a vector mean?” I assume it refers to the autocorrelation of a time series with various lags. It is recommended that the authors focus on the presentation and ensure better clarity in their writing.

2. I find the experiments regarding COP in unconstrained time series generation unclear. I am unsure how COP is applied in such settings and what the specific objective and generative model used are.

3. An ablation study is lacking in the paper, particularly in relation to DiffTime. Although DiffTime is designed based on diffusion models, no diffusion models are included as baselines for comparison. Furthermore, would the performance of GAN baselines be improved with the soft and hard constraints?

**Questions:**

I have a concern regarding the potential optimization issues that may arise when incorporating soft and hard constraints during the training of generative models. I wonder if additional constraints prolong the tuning process or affect the convergence of the models, requiring additional time to train effectively.

**Limitations:**

The presentation and writing require significant improvement.

---

> ### Author Rebuttal · Authors · 2023-08-09
>
> We appreciate the constructive suggestions provided by the reviewer, and we are happy to revise and improve our paper accordingly, incorporating the writing and presentation suggestions pointed out.
>
> ## Weaknesses
>
> **Q1.**
> In particular, we thank the reviewer for pointing out unclear terms and definitions. In the revised paper we will clarify and change any ambiguous terms:
> - By "seed for generation" (line 133) we mean a starting time-series (TS) sample which is modified to generate a new sample (data augmentation);
>  - The "desired property" (line 145) is a function over the original TS data that defines the statistical properties we want to preserve in the generated sample. In line 151-152, we mention that this is a function z($\cdot$) over the TS vector and, as the reviewer correctly supposed, auto-correlation is one example of such a function. We use the term "property" to keep the definition more general, as different TS properties can be specified and preserved. We will improve this section by giving also examples of different properties.
>
> **Q2.**
> We will improve the  COP unconstrained generation section to make it more clear for the reader. The main idea behind the COP formulation for TS generation is the following: we can frame the task of generating a TS sample as optimizing the value of a set of ordered points that make up the TS sample, such that the sample satisfies domain properties (like auto-correlation).
> To do this, we define the domain properties as constraints, and we define the objective as optimizing (maximizing) the distance between the generated sample (set of points) from a starting sample (taken from the dataset, or randomly generated) using the L2 distance between the two TS. With this generation problem setting, we can use existing COP solvers (e.g., SLSQP) to get new samples by solving those non-linear constraints and objectives. Thus the COP solver using the specified constraints and objective function becomes the generative process. There is no additional generative model needed.
>
> In unconstrained TS generation with COP, the objective remains maximizing the difference between an input real TS and the generated TS sample (using L2 distance).  Even when unconstrained -- as in the user has not specified additional constraints-- the COP formulation for TS generation will have inequality constraints associated to preserving domain specific properties, such as auto-correlation, of the starting sample TS. The impact of different input TS (be it real or noise) is investigated in the supplementary material, Section G.3.
> Finally, we would like to highlight that COP has been specifically designed to easily support complex constraints with a straightforward manageable generation procedure, while our diffusion models can provide more sophisticated data generation.
>
> **Q3.**
> We thank the reviewer for pointing out important missing experiments. We extended the ablation study for our diffusion model architecture, and we evaluated the impact of:
> - different number of diffusion steps $T$ (Supplementary material Section G.1);
> - different noise variance schedulers (Supplementary material Section G.2);
> - different model layers and hyper-parameters (discussed below);
>
> We introduce the following variants of DiffTime:
> - *DiffTime-K-Heads* - we change the number of \textit{attention heads}, from 1 to 8;
> - *DiffTime-LSTM* - we replace the convolutional layers using recurrent layers (i.e., LSTM) along the attention mechanism, which is particularly successful for imputation and interpolation of TS[7];
> - *DiffTime-full-LSTM* - we replace all the convolutional and transformer layers by using LSTM layers, which is common for time-series generation[8];
> - *DiffTime-full-CNN* - we replace the transformer layers using convolutional layers;
>
> $
> \begin{array}{l|lll}
> \hline
> \text{Algo} &  \text{Discr-Score} &   \text{Pred-Score}  &   \text{Inference-Time}   \\\\ \hline
> \text{DiffTime-1Heads} & 0.03\pm0.02 &  0.21\pm0.00 & 0.02\pm0.01 \\\\
> \text{DiffTime-4Heads} & 0.06 \pm 0.02 &  0.21\pm0.00 & 0.04\pm0.0 \\\\
> \text{DiffTime-8Heads} & 0.05 \pm 0.03 &  0.21\pm0.00 & 0.02\pm0.01 \\\\
> \text{DiffTime-LSTM}  & 0.06 \pm 0.01 &  0.21\pm0.00 &  0.02\pm0.01 \\\\
> \text{DiffTime-full-LSTM} &   0.50 \pm 0.00 &  0.21\pm0.00 & 0.02\pm0.01 \\\\
> \text{DiffTime-full-CNN}  &   0.14 \pm 0.04 &  0.21\pm0.00 & 0.03\pm0.01 \\\\
> \hline
> \end{array}
> $
>
> The results highlight the performance of the current architecture, which uses transformer layers, where the number of attention heads can be tuned according to the input dataset.
>
> Finally, we would like to highlight that the GAN-based benchmarks have been adapted to deal with soft and hard constraints: we conditioned the architectures on the soft-constraint, while we add a penalty term in the training loss for the hard constraints. The results in the main paper show improved satisfaction-rate (yet lower than those of our approaches), but slightly worst performance for hard-constraints in terms of discr. and pred. scores, and distribution similarity. We found that the GAN-based architectures are inherently unstable and harder to train for hard-constraints.
>
> ## Questions
>
> **Q.1** We empirically noticed that:
> - soft-constraints (which are used to condition the model generation) often make the training faster, but they also increase the chance of over-fitting (for diffusion models) or having mode collapse (for GAN-based models).
> - hard-constraints enforced through a penalty term in the model loss, usually aggravate the training procedure. We experienced more training time, and more instability for GAN-based architectures. However, the training of our Guided-DiffTime is not affected by any constraints, as they can be added at inference time, making its training easier and faster.
>
> **References in the global rebuttal comment**

---

> > ### Comment · Reviewer_DGpF · 2023-08-20
> > **Official Comment by Reviewer DGpF**
> >
> > I appreciate the authors took the time to address my concerns. I now tend to accept the paper.

---

### Official Review · Reviewer_5wsv · 2023-07-07

**Soundness:** 4 excellent
**Presentation:** 4 excellent
**Contribution:** 4 excellent
**Rating:** 8
**Confidence:** 3

**Summary:**

The authors propose multiple methods to solve the constrained time-series generation problem: Given a time-series and a set of (hard or soft, local or global) constraints, the goal is to generate a time-series which is realistic, adheres to the input constraints, yet sufficiently different from the original inputs. The authors propose multiple methods starting from simpler constrained optimization to more complicated diffusion model based solutions relying on recent advances in that space. The authors then compare their models to some state-of-the-art methods and demonstrate that their method's efficacy.

**Strengths:**

I really like this work. It is well-written, motivated and clear.
I am not an expert in diffusion models, but the ideas, the problem formulation and the solution are interesting and make sense to me.
The authors do a good job at evaluating their methods and demonstrate superior performance in comparison to state-of-the-art methods. The authors use standard datasets to evaluate their methods.

**Weaknesses:**

I do not see any weaknesses in the proposed method.

**Questions:**

N/A

**Limitations:**

The authors have not mentioned any limitations of their proposed work. I would encourage them to add a discussion about the same.

---

> ### Author Rebuttal · Authors · 2023-08-09
>
> We thank the reviewer for the positive feedback, and for pointing out an aspect that was not sufficiently discussed. We agree that discussing the limitations of our work will enhance the overall quality of our paper and its impact on the academic community. We will add a more extensive discussion in the final paper, extending the discussion on tradeoffs that we currently have about our approaches.  Finally, we plan to discuss the training and convergence times, as well as any potential limitations can be found in replicating our work.

---

> > ### Comment · Reviewer_5wsv · 2023-08-13
> > **Thanks for the response!**
> >
> > I would like to thank the authors for their work. I stand by my score.

---

### Official Review · Reviewer_U4ao · 2023-07-13

**Soundness:** 3 good
**Presentation:** 3 good
**Contribution:** 3 good
**Rating:** 6
**Confidence:** 4

**Summary:**

In this paper, the authors investigate the problem of time-series generation under different types of conditions. Since the existing methods usually require re-training when the conditions should be changed and the rejection sampling might require heavy computation, the authors use the conditional diffusion model to generate the time-series data with hard and soft constraints. The authors evaluate the proposed method and several datasets and obtain good performance.

**Strengths:**

The problem of time-series generation that the authors investigate is interesting and important. And the authors devise a flexible framework for this problem. Moreover, the proposed method achieves ideal performance and is is environmentally friendly.

**Weaknesses:**

1.	According to equations on Pages 4 and 5, the authors assume that the data points in each timestamp follow the Gaussian distribution and that the time-series data might be stationary. However, some time-series data, e.g., financial data, are influenced by unexpected policies, so the noise or the generation process of the time-series data might be changed, and the data become nonstationary. How can the proposed method address these challenges?
2.	Some implementation details of the proposed method are not clear. For example, what is the architecture of $\epsilon_\theta(x_t, t)$ in equation (4)? Can the authors provide some figures for the proposed method?
3.	According to Algorithm 1, the data are generated via a linear generation process, which might be a strong assumption. Can the proposed method be extended to other more challenging cases?
4.	The compared methods might be too weak and the authors only consider one latest methods. It is suggested that the authors should consider more latest compared methods like [1][2][3].
5.	I think the authors should consider more metrics for the proposed method. For the trend constraint as shown in Figure 1(b), it is suggested that the authors should employ DTW to measure the distance between the original trend and the generated data. Moreover, since the authors consider the sinusoidal data, which is seasonal, it is suggested that the authors should employ spectral transformation methods like Fourier transform and compare the basis.

[1] Deep Latent State Space Models for Time-Series Generation

[2] Generating multivariate time series with COmmon Source CoordInated GAN (COSCI-GAN)

[3] Towards Generating Real-World Time Series Data

**Questions:**

Please refer to the weaknesses.

**Limitations:**

Please refer to the weaknesses.

---

> ### Author Rebuttal · Authors · 2023-08-09
>
> We thank the reviewer for pointing out aspects that were not clearly explained in the paper.
>
> **Q1/3** Our diffusion models use a Markov chain to convert a simple known distribution (i.e., a Gaussian) into the target (data) distribution using a diffusion process[1], without any strong assumptions on the target data distribution. Nevertheless, we agree with authors that time-series can have unexpected changes, and distributional shifts pose new specific modelling challenges[11]. We will investigate such scenarios in future work. Regarding to Algorithm 1 of Guided-DiffTime, we will clarify that we approximate $e_{\theta}$ using a neural network. Moreover is it possible to incorporate recent work to operate the generative process into a latent space [6], to deal with more challenging scenarios. If instead, the reviewer is referring to Algorithm 1 of supplementary material, we will clarify that COP uses an optimizer that supports non-linear constraints (SLSQP optimizer).
>
> **Q2** We have now included an illustration showing the architecture of $e_{\theta}(x_t, t)$ shared across our diffusion models (Figure 1 of PDF rebuttal). We will include such illustration in the main paper.
>
> **Q4** In our current effort, we have integrated *COSCI-GAN*[2] and *RTSGAN[3]* models for unconstrained time-series (TS) generation, as well as for constrained TS generation, specifically focusing on the OHLC constraint. For this constraint, we modified the training procedure by introducing a penalty loss, which penalizes the generative models proportional to how much the generated time-series violate the input constraint. While we have been able to replicate and adapt *RTSGAN*, we note that despite our efforts to replicate *COSCI-GAN* using the provided hyper-parameters and code from the paper, the obtained results are different from those of the original paper. We include preliminary results, and will continue our efforts to integrate such work. Lastly, we were not able to incorporate the work in [4] in our rebuttal due to limited time and difficulty accessing the source code, but plan to include it in the final version.
>
> - *Unconstrained generation (Stock Data)*. We report the quantitative metrics, Discr. and Pred. Score, in following table, while t-SNE analysis is shown in Figure 9 of PDF rebuttal. We also included additional metrics, as kindly suggested by the reviewers. Following the literature on financial TS [5], we analyse the distribution of price returns in Figure 3, and their autocorrelation distribution in Figure 4. *RTSGAN* achieves notable performance in terms of Discr. and Pred. scores, with good distributional similarity from the t-SNE chart in Figure 9. However, with respect to properties pertinent to financial data (Figure 3 and 4), the synthetic data has higher autocorrelation than real data, potentially stemming from multiple GRU layers; and more shallow return distribution (it is well known that asset returns have fat tailed distributions[5]).
>
> $\begin{array}{l|lll}
> \hline
> \text{Algo} &  \text{Discr-Score} & \text{Pred-Score} & \text{Inference-Time} \\\\ \hline
> \text{COP (Ours)} & .050 \pm .017 & .041 \pm .001 & 1.01 \pm 0.00   \\\\
> \text{DiffTime (Ours)} & .097 \pm .016 & .038 \pm .001 & 0.02 \pm 0.00  \\\\
> \text{COSCI-GAN} &  .412 \pm .002 & .088 \pm .000 & 0.00\pm0.00 \\\\
> \text{RTSGAN} & .024\pm.007 & .036\pm.000 & 0.00\pm0.00 \\\\
> \hline
> \end{array}$
>
> - *OHLC Constrained generation (Stock Data)*.  For constrained TS generation, in Figure 10 of PDF rebuttal and in the table below, we can observe similar performance as in the unconstrained setting for *COSCI-GAN* and *RTSGAN*, both in terms of  distributional similarity, discr. and pred. scores. However, looking at the satisfaction rate (i.e., percentage of time-series respecting the input constraint), our methods outperform the two benchmarks. Most importantly, our Guided-DiffTime model stands out for its remarkable capacity to accommodate new constraints without any retraining, constituting a fundamental innovative contribution to the literature on generating TS data.
>
> $\begin{array}{l|llll}
> \hline
> &  \text{Discr-Score} &   \text{Pred-Score} & \text{Inference-Time} & \text{Satisfaction Rate} \\\\ \hline
>   \text{COP (Ours)} & 0.04 \pm 0.02 & 0.04\pm0.00 & 2.17 \pm 0.10 & 1.00\pm0.00  \\\\
>   \text{GuidedDiffTime (Ours)} & 0.08 \pm 0.00 & 0.04\pm0.10 & 0.15 \pm 0.00 &  0.72 \pm 0.02  \\\\
>   \text{LossDiffTime (Ours)} & 0.35 \pm 0.04 & 0.04\pm0.01 & 0.14 \pm 0.00 &  0.69 \pm 0.01  \\\\
>  \text{COSCI-GAN} &  0.45 \pm 0.01 &  0.09 \pm 0.00 & 0.00\pm0.00 & 0.02 \pm 0.00 \\\\
>  \text{RTSGAN} &  0.02\pm0.01 &  0.04\pm0.00 & 0.00\pm0.00 & 0.54 \pm 0.02 \\\\
> \hline
> \end{array}$
>
> **Q5.** Following the reviewer suggestions, we have introduced additional metrics to evaluate the synthetic TS for the sinusoidal trend-constrained generation. In the table below we included distances based on Dynamic Time Warping (DTW) and spectral transformation (L1 distance of Fourier basis). Thanks to the suggested metrics the results better highlight how some methods, like RCGAN, are able to capture the trend, even if shifted, which is also visible on Figure 6 of the supplementary material. Our DiffTime method has the best performance in these measures as well.
>
> $\begin{array}{l|lllll}
> \hline
> & \text{Discr-Score} & \text{Pred-Score} & \text{L2 Distance} & \text{DTW Distance}
> & \text{Fourier-based Distance} \\\\ \hline
> \text{COP (Ours)} & 0.01\pm0.01 & 0.20\pm0.00 & 46.3\pm32.9 & 35.8\pm25.8 & 0.57\pm0.57 \\\\
> \text{DiffTime (Ours)} & 0.01\pm0.01 & 0.20\pm0.00 & 35.57\pm16.99 & 27.57\pm13.12 & 0.49\pm0.57
> \\\\
> \text{GT-GAN} & 0.04 \pm 0.03 & 0.22 \pm 0.00 & 1699.4\pm1253.1 & 1692.5\pm1253.9 & 1.74\pm2.51
> \\\\
> \text{TimeGAN} & 0.02 \pm 0.02 & 0.20\pm0.00 & 121.35\pm61.30 & 87.29\pm50.25 & 1.06\pm1.11
> \\\\
> \text{RCGAN} & 0.02 \pm 0.01 & 0.20\pm0.00 & 124.82\pm83.29 & 95.73\pm72.62 & 0.70\pm0.75
> \\\\ \hline
> \end{array}$
>
> **References in the global rebuttal comment**

---

> > ### Author Response · Authors · 2023-08-15
> > **"Deep Latent State Space Models for Time-Series Generation" - Comparison**
> >
> > Following the reviewer's suggestion, we have successfully obtained the source code for the paper 'Deep Latent State Space Models for Time-Series Generation.' We're now testing it on our datasets and preparing to adapt it for constrained time-series generation.
> >
> > To the best of our knowledge, the suggested work proposes a novel state-of-art method for continuous-time and long-sequences generation. However, it does not address constrained time-series generation. To study its performance in our setting, we plan the following straightforward changes: - we introduce conditional variables (for trend constraint); - we employ a penalty term to regularize the decoder output and improve the constraint satisfaction (of hard constraints). However, we note that adding such penalty terms may complicate the model training, and, most important, re-training is required when the constraints change. Conversely, our Guided-DiffTime incorporates constraints at inference without requiring specific (re)training steps.
> >
> > We thank the reviewer again for the suggestion. We will provide an extended related work in the supplementary material, mentioning the main differences between our work and suggested methods.

---

### Author Rebuttal · Authors · 2023-08-09

We would like to thank all the Reviewers for their constructive and valuable comments. We revised our work according to the reviewer's comments, and new results are included in this rebuttal, and attached PDF. In particular:

$\bullet$ We included 2 new state-of-art benchmarks for time-series generation;

$\bullet$ We introduced 3 baseline approaches to synthetic time-series generation;

$\bullet$ We included more metrics to evaluate the generated synthetic time-series and their statistical properties;

$\bullet$ We added more details on our diffusion model architecture, including a graphical representation;

$\bullet$ We performed a more detailed ablation study on the diffusion model architecture;

$\bullet$ We improved the study of COP approach, showing new experiments and its fine-tuning capabilities in more detail. By using COP for fine-tuning, after any generative model, we observe that we can double the constraint-satisfaction rate of generated time-series, while keeping the same distributional properties (see Figure 6 and Figure 7 of PDF rebuttal);

$\bullet$ We provided more discussion and examples on how constrained time-series generation can be helpful in real-world problems;

$\bullet$ We are incorporating the writing and presentation suggestions pointed out by reviewers in the final paper version.

We are happy to discuss our work further, and we believe our manuscript has improved by incorporating reviewer suggestions, providing more valuable insights and results. We will integrate the new developments in the final paper version. A more detailed discussion is provided in each rebuttal.

**Novelty:**
Finally, we would like to highlight that our approach provides realistic time-series generations for both constrained and unconstrained generation. In fact, while existing GAN-based architectures have comparable performance in some unconstrained generative scenarios, they are not directly suitable for constrained generation. Modifying the GAN loss function to incorporate constraints is not trivial, and can aggravate the inherent instability of GANs, and rejection-sampling is expensive for complex constraints. Instead, our work directly targets constrained generation. Our Guided-DiffTime can generate different constrained time-series, without any special (re)training, as constraints can just be introduced at inference time. This drastically reduces the computational cost and carbon footprint (as empirically shown in Table 6 of Supplementary material). Additionally, our COP approach can be used instead of rejection sampling to revise synthetic samples from any generative model.



**All references of the rebuttal comments:**

[1] - J. Sohl-Dickstein, et al. "Deep Unsupervised Learning using Nonequilibrium Thermodynamics" ICML 2015.

[2] - A. Seyfi, et al.. "Generating multivariate time series with COmmon Source CoordInated GAN (COSCI-GAN)." NeurIPS 2022

[3] - H. Pei, et al. "Towards generating real-world time series data." ICDM 2021

[4] - L. Zhou, et al. "Deep latent state space models for time-series generation." ICML 2023.

[5] - JP. Bouchaud,  et al. Trades, quotes and prices: financial markets under the microscope. Cambridge University Press, 2018.

[6] - R. Rombach, et al. "High-resolution image synthesis with latent diffusion models." IEEE/CVF 2022.

[7] - S.N. Shukla et al. "Multi-Time Attention Networks for Irregularly Sampled Time Series." ICLR 2020.

[8] - Mogren, Olof. "C-RNN-GAN: Continuous recurrent neural networks with adversarial training."

[9]  - Kirilenko, Andrei, et al. "The flash crash: High‐frequency trading in an electronic market." The Journal of Finance 72.3 (2017): 967-998.

[10] - Mazur, Mieszko, Man Dang, and Miguel Vega. "COVID-19 and the march 2020 stock market crash. Evidence from S\&P1500." Finance research letters 38 (2021): 101690.

[11] - J. Jia, A. Benson. Neural Jump Stochastic Differential Equations. Neurips 2019.

[12] - D. Cao, et al. "DSLOB: a synthetic limit order book dataset for benchmarking forecasting algorithms under distributional shift"

[13] - S. Bartram,et al. "Machine learning for active portfolio management." The Journal of Financial Data Science (2021)

[14] - F. Ferreira "Artificial intelligence applied to stock market trading: a review." IEEE Access (2021)

---

### Decision · Program_Chairs · 2023-09-21

**Decision:**

Accept (poster)

**Comment:**

This paper solves the problem of constrained time series generation problem, where several types of constrained e.g., hard and soft, local and global, are taken into account. The authors achieve the ideal performance and illustrate the effectiveness of the proposed method. After rebuttal, the authors address most of the problems of the reviewers. Some reviewers consider that this paper is well-written. The decision is accepted.